# Viscoelastic Effects during Tangential Contact Analyzed by a Novel Finite Element Approach with Embedded Interface Profiles

**Jacopo Bonari** *,† and **Marco Paggi** †

IMT School for Advanced Studies Lucca, Piazza San Francesco 19, 55100 Lucca, Italy; marco.paggi@imtlucca.it

* Correspondence: jacopo.bonari@imtlucca.it
† These authors contributed equally to this work.

**Abstract:** A computational approach that is based on interface finite elements with eMbedded Profiles for Joint Roughness (MPJR) is exploited in order to study the viscoelastic contact problems with any complex shape of the indenting profiles. The MPJR finite elements, previously developed for partial slip contact problems, are herein further generalized in order to deal with finite sliding displacements. The approach is applied to a case study concerning a periodic contact problem between a sinusoidal profile and a viscoelastic layer of finite thickness. In particular, the effect of using three different rheological models that are based on Prony series (with one, two, or three arms) to approximate the viscoelastic behaviour of a real polymer is investigated. The method allows for predicting the whole transient regime during the normal contact problem and the subsequent sliding scenario from full stick to full slip, and then up to gross sliding. The effects of the viscoelastic model approximation and of the sliding velocities are carefully investigated. The proposed approach aims at tackling a class of problems that are difficult to address with other methods, which include the possibility of analysing indenters of generic profile, the capability of simulating partial slip and gross slip due to finite slidings, and, finally, the possibility of simultaneously investigating dissipative phenomena, like viscoelastic dissipation and energy losses due to interface friction.

**Keywords:** viscoelasticity; contact mechanics; finite element method

## 1. Introduction

A recently developed finite element procedure is herein extended and applied to the analysis of the transient and steady state sliding of a rigid indenter over a deformable material. In accordance with the requirements of current industrial applications, which demand increasingly complex contacting topologies, often down to the micro-scale, together with the analysis of concurrent interface phenomena, like friction and wear, it is shown that the present approach is capable of dealing with arbitrarily complex surfaces and, thanks to the flexibility of the finite element method, to account for any kind of material law.

Indeed, real viscoelastic materials present a time-dependent mechanical response that varies across several orders of magnitude of time and intensity. Therefore, a simple model with a linear Hookean spring in series with a single Newtonian dashpot is far from being representative. For instance, for Ethylene Vynil Acetate (EVA) used as an encapsulating material for photovoltaics, a power-law decay of the Young's modulus with time has been reported [1,2], which can be well-modelled by a fractional viscoelastic model [3–5] as a limit of a Prony series representation with several arms. Its approximation for engineering applications usually requires the use of at least three arms in the Prony series, in order to provide meaningful stress analysis predictions.

In this study, we propose an extension of the variational approach that is based on the interface finite element with eMbedded Profile for Joint Roughness (MPJR) recently proposed in [6,7] for frictionless normal contact problems, and further generalized in [8] in order to simulate frictional partial slip scenarios, to accommodate also finite interface sliding displacements. The methodology, which allows embedding any contact profile as an exact analytical function into an interface finite element, overcomes the cumbersome procedure required by standard finite element methodologies to explicitly discretize the geometry of the boundary exposed to contact. In the MPJR method, the boundary is treated as flat and its actual perturbation from flatness is included as a correction to the normal gap. Since the MPJR method is set to operate within the finite element method (FEM), it presents all the advantages of FEM to solve linear and nonlinear boundary value problems with any arbitrary material constitutive law and structural geometry.

A representative contact problem involving a rigid indenter with harmonic profile acting over a viscoelastic layer of finite depth, perfectly bonded to a rigid substrate, is addressed in order to demonstrate the capabilities of the proposed approach. The loading history will include an applied displacement normal to the contacting interface during a first stage, with a progressive increase in the contact area. Afterwards, the normal displacement is held constant and a horizontal far-field displacement in the sliding direction is applied, in order to simulate the stick-slip transition and then the steady-state sliding regime. Friction is considered along the interface and it is mathematically treated with a regularized Coulomb frictional law. Different sliding velocities, which are relevant for the behaviour of a viscoelastic material, are examined. Numerical simulations provide useful insight into the distribution of the tangential tractions in all of the phases of the sliding process. When considering different Prony series representations with a number of arms varying from one to three, the computational approach allows for quantifying the effect of refining the viscoelastic constitutive model by introducing additional relaxation times.

## 2. Materials and Methods

### 2.1. Proposed Solution Scheme for the Contact Problem

In order to investigate the effect of different viscoelastic models along with frictional effects, the contact problem involving a rigid indenter that is characterised by a harmonic profile acting over a layer made of a linear viscoelastic material is addressed. Here, is important to remark that there are no restrictions on the shape of the indenting profile, which can be chosen as an analytical function, or it can be provided as a discrete set of elevations. In the latter case, an external file provided by a profilometer, with a simple two-columns data structure with sampling point coordinate and its elevation, can be used in input. To use such data, one has to keep in mind that the boundary has to be discretized by using MPJR interface finite elements with a uniform spacing dictated by the profilometer resolution, to achieve a one-to-one correspondence between finite element nodes and profilometer sampling points. The assignment of the elevation to each finite element node can be efficiently done only once, just at the beginning of the simulation, by a simple searching algorithm looking for the global coordinate of the finite element node that matches the coordinate stored in the external data file. Subsequently, elevations are efficiently stored in a history variable, in order to avoid multiple reading from external files during the Newton–Raphson iterations and in the next loading steps of the simulation. Further details on the finite element procedure can be found in [6].

### 2.2. MPJR Formulation

For the solution of the contact problem, the MPJR interface finite element that is exposed in [6,8] is employed. It consists in a 4-nodes, zero-thickness element mutuated by the Cohesive Zone Model (CZM) and used in the context of nonlinear fracture mechanics. The framework is applied to the problem of a rigid body with a complex boundary making contact with another deformable body characterised by a smooth interface. The core of the approach is re-casting the original geometry of

the problem into a simpler one, consisting only in the deformable bulk and a single layer of interface elements disposed at its boundary, where contact is supposed to take place, as in Figure 1. The actual shape of the indenter is stored nodal-wise in each interface finite element employed for the interface discretization, and it is used to correct the normal gap function that is computed from the flat–flat configuration, in order to account for the exact geometry. This requires a preliminary step, which consists in mapping the indenter profile elevation in correspondence to the right node of the boundary. If the profile has analytic expression, this can be done right at the finite element level exploiting the global coordinates, otherwise the elevation field can be stored in a proper history variable and every entry associated with the correct node. It has to be remarked that, in spite of the present formulation being 2*D*, the proposed framework can be extended to 3*D* problems, provided that, for example, a 8-nodes interface element is used to discretize a surface, instead of a profile, equipped with a suitable friction law.

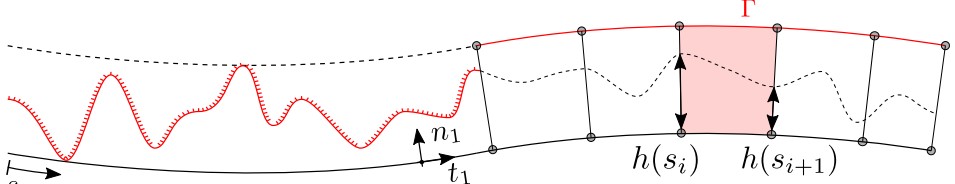

**Figure 1.** Profile discretization and equivalent interface definition. The interface element Γ is defined with the lower two nodes that belong to the deformable bulk, and the others placed at a given offset normal to the lower boundary. An abscissa *s* can be defined along the boundary to map the indenter's elevation field, which is stored inside the element and it is used to correct the normal gap.

Figure 2 shows the kinematics of the element. A vector of unknown nodal displacements $\mathbf{u} = [u_1, v_1, \ldots, u_4, v_4]^T$ is introduced for the evaluation of the tangential and normal gaps, collected in the vector $\mathbf{g} = [g_x, g_z]^T$, which reads:

$$\mathbf{g} = \mathbf{QNLu},\tag{1}$$

where **L** is a linear operator for computing the relative displacements across the interface, **N** is the shape functions matrix, and **Q** is a rotation matrix for transforming displacements from the global to the local reference frame of the element defined by the unit vectors *n* and *t*. The original geometry can be restored with a suitable correction of the normal gap, in the form:

$$\mathbf{g}^* = \begin{bmatrix} g_x \\ g_z + h(\xi, t) \end{bmatrix},\tag{2}$$

where $h(\xi, t)$ maps the profile's shape and position in time. With respect to the formulation that is presented in [8], here the profile shape has been made time-dependent, in order to also account for finite sliding of the rigid indenter. For example, in the case of a flat interface, the result of the indenter sliding with a given constant velocity $v_0$ can be achieved by setting $h(\xi, t) = h(\xi - v_0 t)$.

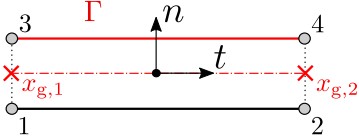

**Figure 2.** Four-nodes, zero-thickness eMbedded Profiles for Joint Roughness (MPJR) interface finite element.

The current value of $t$ is stored at the interface element level while using a time history variable, and it is updated every time step. A standard penalty approach is used in order to enforce the normal contact constraint, leading to

$$p_z = \begin{cases} \alpha g_z^*, \text{ if } g_z^* < 0, \\ 0, \text{ if } g_z^* \geq 0, \end{cases} \tag{3}$$

where $\alpha$ is the penalty parameter.

To deal with coupled frictional problems, a regularized Coulomb friction law [9] is used to set the interface constitutive equation in the tangential direction:

$$q_x = f p_z \tanh\left(\frac{\dot{g}_x}{\dot{\varepsilon}}\right), \tag{4}$$

where $q_x$ is the tangential traction and $\dot{g}_x$ is the sliding velocity, as given by the difference between the velocity of the indenter and the horizontal velocity of the corresponding node. Finally, $\dot{\varepsilon}$ is a parameter governing the slope of the regularised friction law.

The contribution of a single interface finite element to the variational formulation of the bulk material is expressed by its variation in terms of density of energy content, integrated over the domain $\Gamma_e$ that denotes the element itself:

$$\delta\Pi_e = \int_{\Gamma_e} \delta\mathbf{g}^*(\mathbf{v})^T \mathbf{p}(\mathbf{u}, \dot{\mathbf{u}}) \, d\Gamma_e, \tag{5}$$

where $\mathbf{v}$ is the virtual displacement field, and the vector $\mathbf{p}$ collects the normal and tangential tractions. As a final step, the variation can be expanded and the integral set to zero, leaving the expression of the nonlinear residual vector, which reads:

$$\mathbf{R}_e(\mathbf{u}, \dot{\mathbf{u}}) = \int_{\Gamma_e} \mathbf{L}^T \mathbf{N}^T \mathbf{Q}^T \mathbf{p}(\mathbf{u}, \dot{\mathbf{u}}) \, d\Gamma_e = \mathbf{0}. \tag{6}$$

Because of the nonlinearity of $\mathbf{R}_e$, the Newton–Raphson iterative method has been applied, together with a backward Euler method for time integration.

*2.3. Rheological Model*

Three different Prony series models with a number of arms increasing from one to three are examined in order to assess the effect of viscoelasticity modelling on the overall contact mechanical response. The general equation for the shear relaxation modulus reads:

$$\frac{G(t)}{G^\infty} = \mu_0 + \sum_{n=1}^{3} \mu_n \exp\left(-\frac{t}{\tau_n}\right), \tag{7}$$

where $G^\infty$ is the instantaneous shear modulus (evaluated at $t = 0$), $\mu_n$ are the relaxation coefficients, and $\tau_n$ are the corresponding relaxation times. Equation (7) has been tuned to fit the experimental values of EVA [4]. The model parameters for 1, 2 and 3 arms are collected in Table 1.

**Table 1.** Rheological parameters for Ethylene Vynil Acetate (EVA), where $n$ is the number of Prony series' arms.

| $n$ [–] | $G^\infty$ [Pa] | $\mu_0$ [–] | $\mu_n$ [–] | $\tau_n$ [s] |
|---|---|---|---|---|
| 1 | 568.498 | 0.421 | 0.579 | 0.817 |
| 2 | 674.606 | 0.306 | 0.398 | 0.212 |
|   |         |       | 0.296 | 2.458 |
| 3 | 749.386 | 0.254 | 0.310 | 0.102 |
|   |         |       | 0.226 | 0.545 |
|   |         |       | 0.210 | 4.104 |

The identification of the above parameters has been carried out through a regression over the experimental data that were acquired in the time range $t = 10^{[-1,\dots,+1]}$ s. The following approach has been pursued in order to attain a high degree of accuracy. Firstly, trial relaxation times have been set and a preliminary linear regression has been performed involving $G^\infty$ and $\mu_i$ only. The objective function to be minimised reads:

$$\Pi(\mathbf{x}) = \sum_{k=1}^{N} \left(\mathbf{g}_k \cdot \mathbf{x} - G_k\right)^2, \tag{8}$$

where, for the three arms model, $\mathbf{g}_k = \left[1, e^{(-t_k/\tau_1)}, \dots, e^{(-t_k/\tau_3)}\right]$, $G_k$ is the value of the objective function at the sampling point and $N$ is the number of samplings. The global minimiser $\mathbf{x}^* = \arg\min_x \Pi(\mathbf{x})$ is evaluated and the values of the constants $\mu_i$ and $G^\infty$ are obtained according to:

$$G^\infty \begin{bmatrix} \mu_0 \\ \mu_1 \\ \mu_2 \\ \mu_3 \end{bmatrix} = \mathbf{x}^*, \tag{9}$$

together with the condition $\sum_i \mu_i = 1$, related to the shear modulus at $t = 0$. The obtained coefficients, together with their respective relaxation times, have been used in order to define a vector of guess values $\mathbf{x}_0$ for a second nonlinear regression, in which the relaxation times were also included in the optimisation vector $\mathbf{x}$. The problem has been solved iteratively, updating the starting vector $\mathbf{x}_0$ every cycle using the results that were obtained in the previous. Convergence is achieved within 5 iterations, when considering a relative error that is given by $(\mathbf{x}^* - \mathbf{x}_0)/\mathbf{x}_0$ and a tolerance $\varepsilon = 10^{-15}$. This procedure has also been repeated in the same way for the 1 and 2 arms models.

Once the parameters are identified, the Young's relaxation modulus $E(t)$ can be obtained from $G(t)$, and the behaviour of the three models can be investigated in time and frequency domains. The analysis in the frequency domain can be performed by defining a complex modulus $\hat{E}(\omega)$, obtained via a Fourier transform of $E(t)$, which can be expressed as:

$$\frac{\hat{E}(\omega)}{E^\infty} = \mu_0 + \sum_{i=1}^{n} \mu_i \frac{\tau_i^2 \omega^2}{1 + \tau_i^2 \omega^2} + \imath \sum_{i=1}^{n} \mu_i \frac{\tau_i \omega}{1 + \tau_i^2 \omega^2}. \tag{10}$$

In the expression above, $\imath$ denotes the imaginary unit and the index $k$ defines the number of arms being considered. It can be easily noticed that, for the single arm model, the maximum viscoelastic effect manifests in correspondence to the critical excitation frequency $\omega^\star = \sqrt{\mu_0}/\tau_1$.

Figure 3a shows the plot of $E(t)$. Figure 3b,c show the values of the loss modulus and the storage modulus, which were obtained as the imaginary part $\Im\hat{E}(\omega)$ and the real part $\Re\hat{E}(\omega)$ of the complex modulus $\hat{E}(\omega)$, respectively. Finally, Figure 3d shows the loss tangent, given as the ratio of

the imaginary part over the real part. As a comparison, the same quantities are also plotted for the relaxation modulus obtained for a model that is based on fractional calculus, which reads:

$$E_{\mathrm{f}}(t) = \frac{E_{\mathrm{f},\alpha} t^{-\alpha}}{\Gamma(1-\alpha)}. \tag{11}$$

In Equation (11), $E_{\mathrm{f},\alpha} = 814.7\,\mathrm{Pa\,s}^{\alpha}$ and $\alpha = 0.226$ have been chosen in order to fit the experimental data in [4], being $\Gamma(\cdot)$ the gamma function.

The simulation of the power-law viscoelastic response seen in the experiments, which is well approximated by the fractional calculus model, is progressively improved by increasing the number of terms in the Prony series representation. It has to be remarked that, since the Fourier transform of a power-law is a power-law itself, both loss and storage modulus in the frequency domain are represented, on a logarithmic scale, as straight lines.

Their trend can be satisfactory modelled with Prony series only for a narrow band of the whole spectrum, based on the relaxation time(s) employed. Therefore, the relaxation times entering Prony series have to be regarded as design parameters, to be chosen based on the loading history experienced by the viscoelastic material, rather than material parameters. With the values that were chosen here, an accurate estimation of the material response can be expected, at most, over two orders of magnitude, centred on a frequency of 1 Hz.

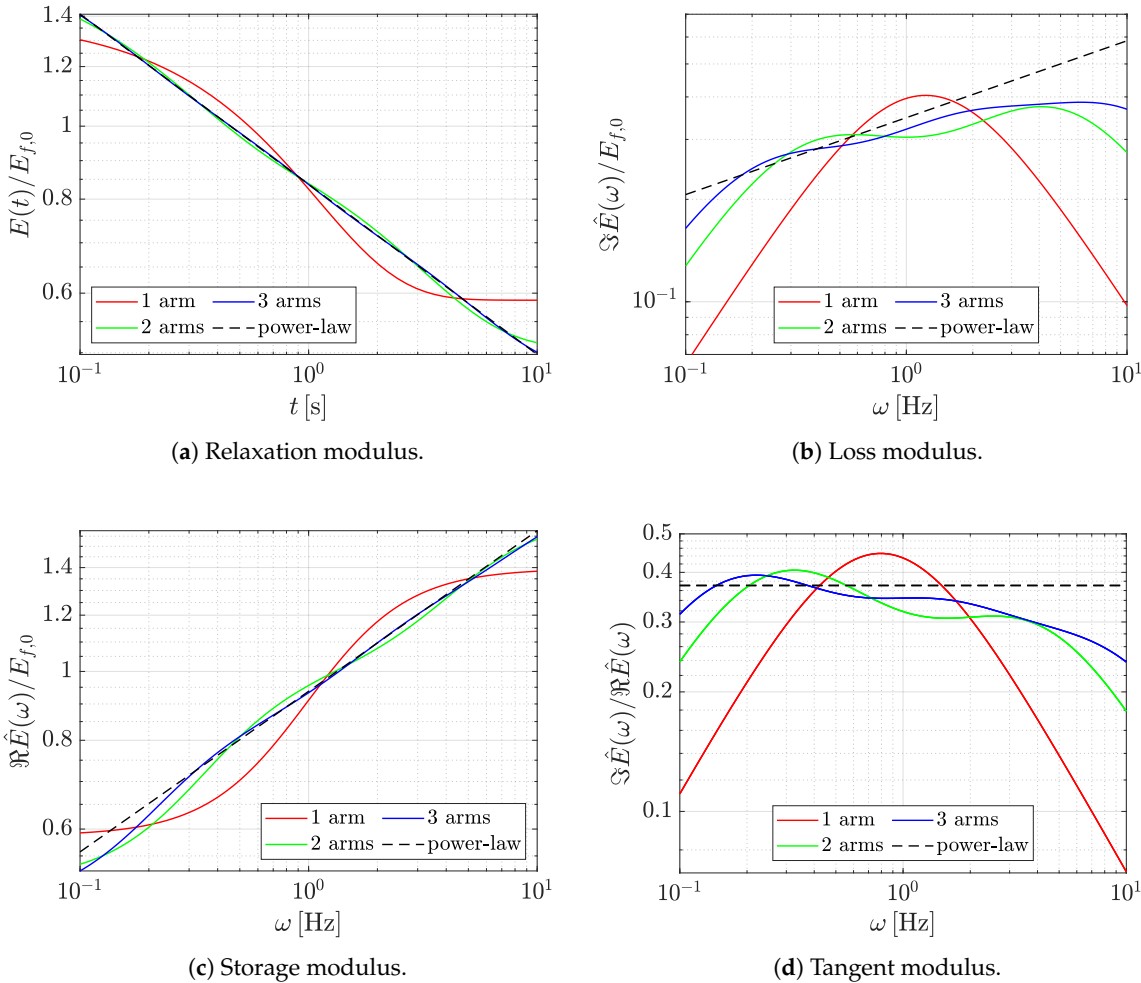

(a) Relaxation modulus.

(b) Loss modulus.

(c) Storage modulus.

(d) Tangent modulus.

**Figure 3.** Relaxation modulus in time and frequency domain.

## 2.4. Problem Set Up

We focus our attention onto a displacement controlled problem under plane strain assumptions in order to highlight the capability of the proposed approach. In the first stage, a displacement linearly increasing with time is applied along the direction normal to the finite layer, up to a given final value of $\Delta_{z,0} = 2g_0$, reached at time $t = t_0$, which is then held constant. At this point, a tangential displacement with a constant horizontal velocity is applied to the indenter, which starts sliding. The indenter profile is analytically expressed by:

$$\frac{h(x,t)}{g_0} = 1 - \cos\left[\frac{2\pi}{\lambda_0}(x - vt)\right] \tag{12}$$

While the velocity of the application of normal load is the same for all the simulations, and assumed to be quasi-static, for what concerns the horizontal load different sliding velocities have been considered in the range $v_i = 10^{(i-10)/3}[\text{m/s}]$, $i = [1, \ldots, 10]$, with their numerical value being summarised in Table 2.

**Table 2.** Range of horizontal velocities employed.

| $v$ [m/s] |
| --- |
| $1.000 \times 10^{-03}$ |
| $2.154 \times 10^{-03}$ |
| $4.642 \times 10^{-03}$ |
| $1.000 \times 10^{-02}$ |
| $2.154 \times 10^{-02}$ |
| $4.642 \times 10^{-02}$ |
| $1.000 \times 10^{-01}$ |
| $2.154 \times 10^{-01}$ |
| $4.642 \times 10^{-01}$ |
| $1.000 \times 10^{+00}$ |

A regularized Coulomb frictional law [8] is considered, with $f = 0.2$ being the friction coefficient. Figure 4 lists the remaining geometric parameters that describe the problem set, together with the rheological model that is employed for modelling viscoelasticity, which has already been thoroughly discussed in Section 2.3: three different simulations are performed, each of them characterised by one, two, or three terms of a Prony series used for modelling a linear viscoelastic material. The model geometry and applied velocities are the same in all of the cases considered. Finally, periodic boundary conditions have been introduced in correspondence of the two vertical sides of the domain, in order to simulate a semi-indefinite contact in the horizontal direction. The simulations have been performed using the Finite Element Analysis Program FEAP [10], where the MPJR formulation has been implemented as a user element routine. The validation of the proposed computational method is provided in Appendix A.

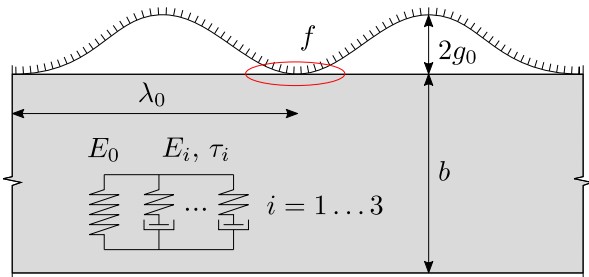

**Figure 4.** Sketch of the model, $b = 1$, $\lambda_0 = b$, $g_0 = 5 \times 10^{-4}\lambda_0$.

## 3. Results

### 3.1. Bulk Stresses

Figure 5 shows the results of FEM simulations for the boundary value problem shown in Figure 4. They refer to the single arm model, but, from a qualitative point of view, the considerations that are going to be drawn below for the bulk stresses also apply to the other two models herein considered.

Figure 5a,b display the stresses developing in the bulk at the end of the normal loading stage, and they display three distinct areas with high stresses where the harmonic profile comes into contact. Because of the presence of friction and, since coupling effect are fully included, an anti-symmetric distribution of $\tau_{xz}$ arises, even in the pure normal loading stage, see Figure 5b. The following two figures represent the same quantities at a subsequent load stage, where the normal imposed displacement has reached its maximum, and the indenter slides at constant velocity. Figure 5c,d show the stresses during the next stage of sliding, corresponding to a lateral shift of the harmonic profile of about half of its wavelength. The advantage of the finite element method is evident from the possibility to consider any finite-size problem geometry and boundary conditions, like, in this case, the output automatically including not only contact tractions, but also bulk stresses.

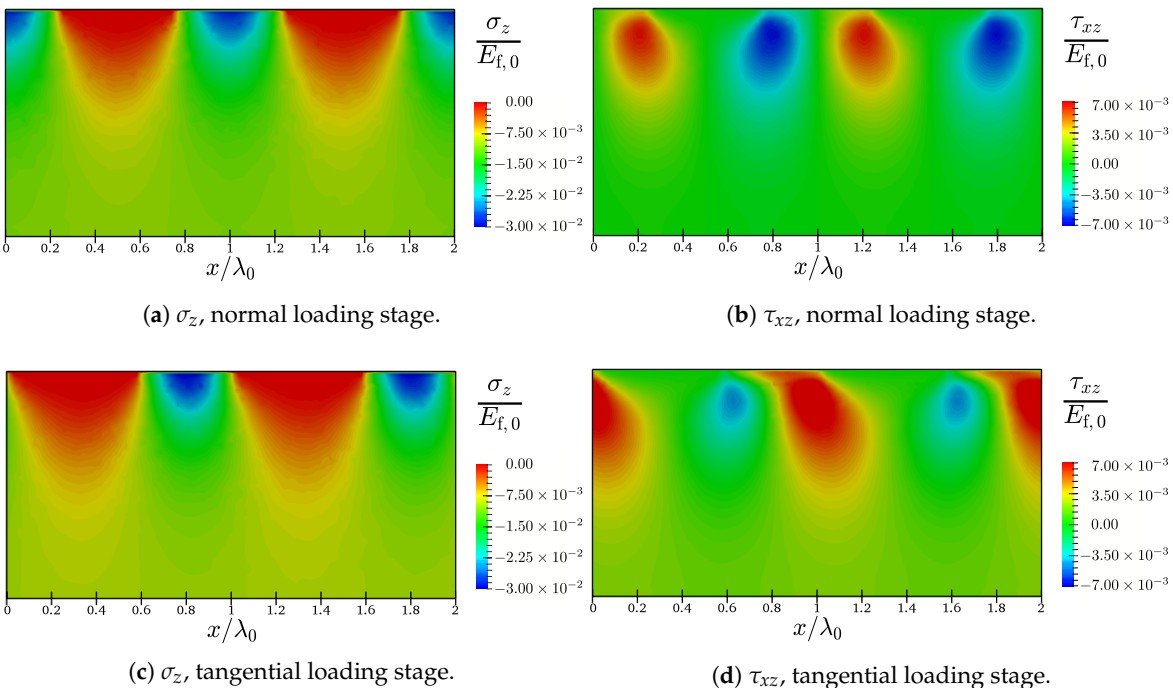

(**a**) $\sigma_z$, normal loading stage.

(**b**) $\tau_{xz}$, normal loading stage.

(**c**) $\sigma_z$, tangential loading stage.

(**d**) $\tau_{xz}$, tangential loading stage.

**Figure 5.** Model predictions: bulk stresses during the normal approach, (**a**,**b**), and during full sliding, (**c**,**d**), all scaled by a reference elastic modulus $E_{f,0} = 8.147 \times 10^2 \, \text{Pa}$.

### 3.2. Interface Tractions

Figure 6 highlights the evolution of contact tractions in time for the single arm model and selected stages of the contact simulation. The curves in Figure 6a correspond to the purely normal loading sequence, where normal contact tractions progressively increase along with the value of the applied normal displacement, which linearly rises from zero up to the final value of $2g_0$. Black curves denote the symmetric distribution of normal contact tractions $p_z(x)$ divided by $E_{f,0}$, while red curves represent the anti-symmetric distribution of tangential contact tractions $q_x(x)$, scaled by $f E_{f,0}$. Points along the interface, where $\|q_x(x)\|/(f E_{f,0})$ equals $p_z(x)/E_{f,0}$, are in a state of slip, while, when the inequality $\|q_x(x)\|/(f E_{f,0}) < p_z(x)/E_{f,0}$ holds, then there is a state of stick.

Figure 6b refers to the next stage of the contact problem when, keeping the normal displacement constant, a far-field displacement linearly increasing with time is applied in the tangential direction. While, for the given rheological model, the results that are shown in Figure 6a are evaluated in a condition of zero tangential velocity, Figure 6b–d are referred to $v = 2.154 \times 10^{-2}$ m/s. This specific value has been chosen amid the other entries of Table 2, because it is in the middle of the range, determining the highest viscoelastic effects, and it is also low enough for analysing the transition from *stick/slip* to *full sliding*, Figure 6b. Here, tangential traction distributions change their shape from the classical anti-symmetric form towards a state of increasing slip, which terminates in the full slip condition. The transition from *stick-slip* to *full slip* is strongly affected by the velocity of the horizontal displacement: the faster the slip, the more abrupt such a transition.

Figure 6c refers to the situation of sliding after full slip (*gross sliding*) and, in particular, it shows the evolution over time of the normal contact tractions. We see a transition from the symmetric contact traction distribution along the whole interface at the onset of full slip, as shown in black, towards other distributions in different scales of grey shifted along the interface to the right, as long as the tangential displacement increases. A certain degree of relaxation is observed after the onset of full slip. As the sliding proceeds in time, virgin material is perturbed, and a recovery in stiffness takes place.

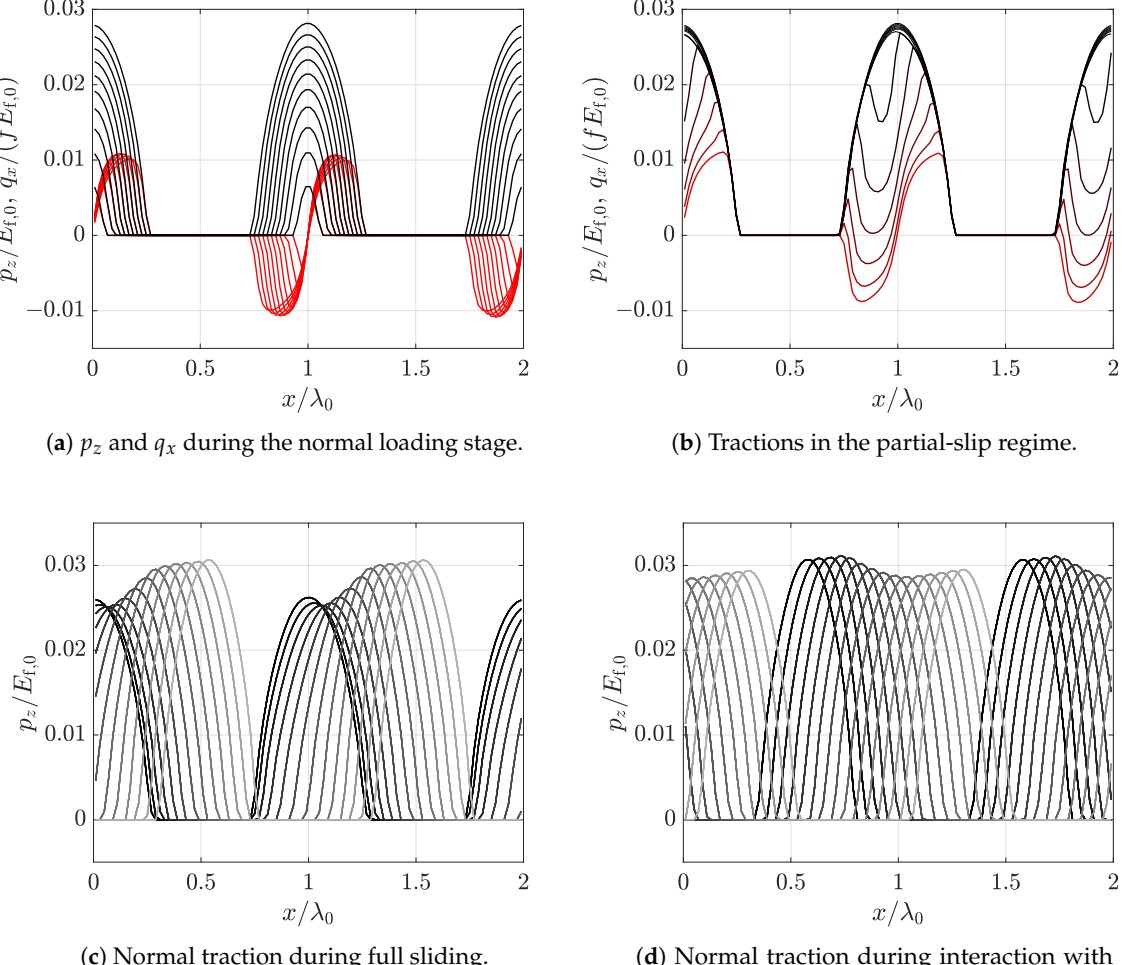

(**a**) $p_z$ and $q_x$ during the normal loading stage.

(**b**) Tractions in the partial-slip regime.

(**c**) Normal traction during full sliding.

(**d**) Normal traction during interaction with an already stressed portion of the interface.

**Figure 6.** Selected distributions of normal and tangential contact tractions during the different stages of loading.

Finally, Figure 6d captures the first overlapping of a new contact zone with a previously loaded portion of the interface. Here, the role of the relaxation time is important, since viscoelastic effects do alter the solution that corresponds to a linear elastic material that has no memory effects.

The resultant tangential force $Q_x$, integral of tangential contact tractions along the interface, is plotted vs. time in Figure 7a–c for the three viscoelastic models investigated herein. In each subfigure, different curves correspond to different far-field horizontal displacement velocities. Darker curves correspond to slower velocities.

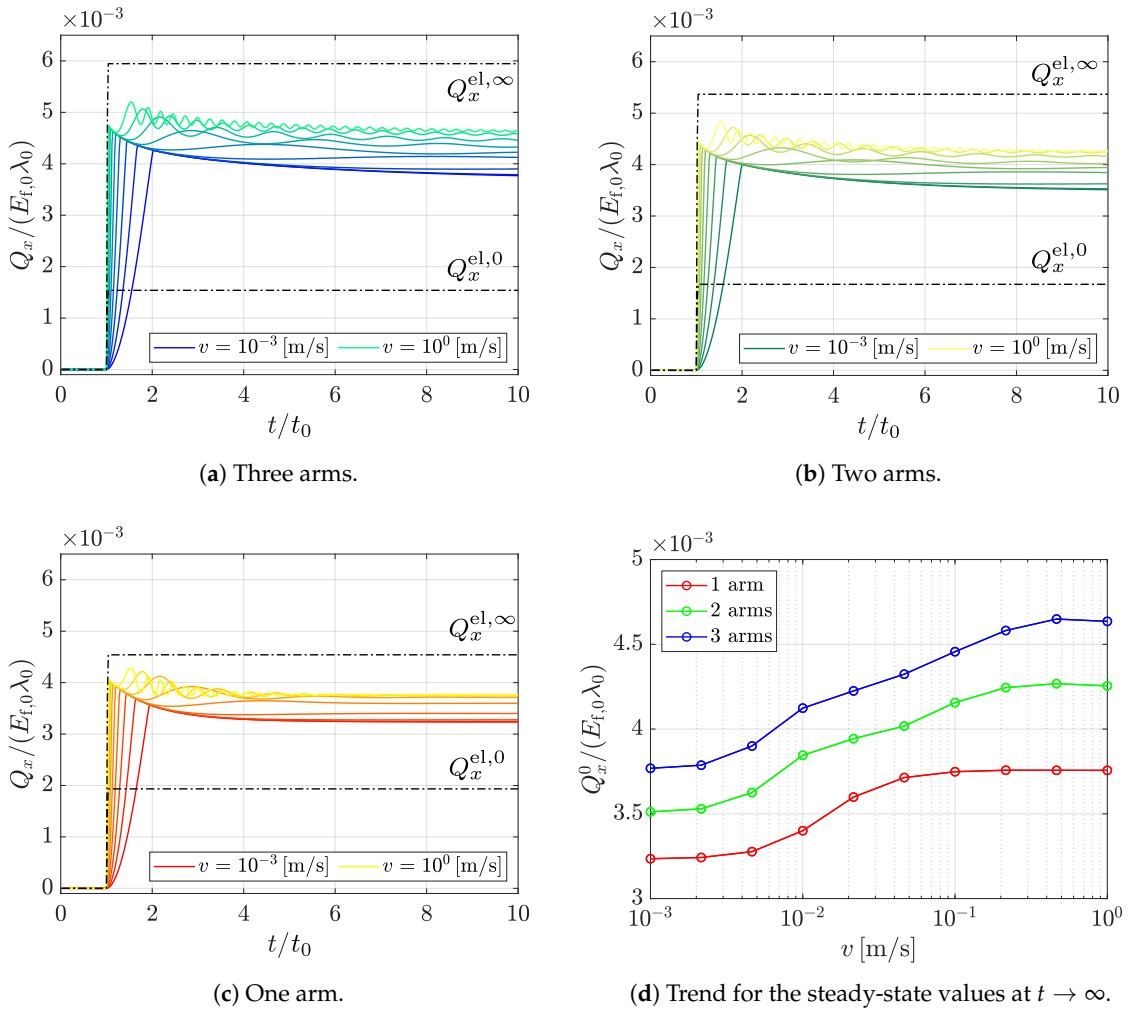

(**a**) Three arms.

(**b**) Two arms.

(**c**) One arm.

(**d**) Trend for the steady-state values at $t \to \infty$.

**Figure 7.** Time evolution of the resultant tangential force $Q_x$ for different rheological models.

In all of the cases, for $t/t_0 \leq 1$, tangential tractions are vanishing, since, in that stage, the imposed displacement is only acting in the normal direction. Therefore, tangential contact tractions are due to frictional coupling effects and their sum over the whole contact zones is vanishing by definition, since they correspond to self-equilibrated distributions. For $t/t_0 > 1$, the indenter starts sliding and we assist to a transition from *stick-slip* to *full slip* with an oscillatory behaviour when the contact profile enters in contact with unrelaxed material portions. When the velocity is low, no rate effects are evident, and the mechanical response is smooth. On the other hand, by increasing the applied velocity, the importance of viscoelasticity increases and oscillating responses do appear.

The integral of tangential tractions related to two linear elastic models that are characterised by short and long term modulus are also plotted in Figure 7; for comparison, see black dash-dotted lines. The elastic moduli are evaluated as:

$$E^{\text{el},\infty} = \lim_{t \to 0} E(t) = E^{\infty} \qquad\qquad E^{\text{el},0} = \lim_{t \to \infty} E(t) = E^{\infty}\left(1 - \sum_{i=1}^{n} \mu_i\right) \qquad (13)$$

The curves $Q_x^{\text{el},\infty}$ and $Q_x^{\text{el},0}$ are evaluated under the assumption of linear elasticity, neglecting the dynamic effects. For this reason, they lead to constant values as soon as the horizontal far-field displacement is applied, without any oscillation. The only factor that plays a role is the velocity, which governs the transition from *stick/slip* to full sliding. In the figures, only the curves that correspond to the highest value of $v$ are plotted. In all three models, the instantaneous (higher) and long term (lower) curves are extreme bounds to the values that are related to viscoelastic simulations, with a gap increasing from the single arm to the three arms model, consistent with their respective stiffness.

The steady-state solution strongly depends on the rheological properties of the material, as shown in Figure 7d. In general, for the present case study, the higher the number of arms, the higher the total tangential force. In all cases, the highest velocity determines the highest value of the steady state $Q_x^0$. This is in accordance with the fact that, in a condition of *gross slip*, $Q_x = f N_z$, and for high velocities, the material is excited in its high frequency region, thus resulting in a vertical response that is governed by the higher *glassy* Young's modulus. The increased stiffness leads to higher $N_z^0$ and, in turn, higher $Q_x^0$ values.

## 4. Conclusions

In this study, a novel finite element procedure has been proposed, which allows for investigating transient and steady state sliding of a rigid indenter over a viscoelastic continuum. In particular, the representative problem of an indenter with harmonic profile sliding over a viscoelastic layer of finite depth has been analysed, employing different sliding velocities together with three different rheological models, which are characterised by Prony series with one, two, and three arms, respectively. A regularised version of the classic Coulomb friction law has been employed for the evaluation of the interface tangential tractions.

Numerical results pinpoint a strong dependence of the mechanical response in terms of steady-state forces $N_z$ and $Q_x$ on both the velocity and rheological model employed, obtaining increasing forces for higher velocities and more relaxation terms that are involved in the rheological approximation.

It is worth mentioning that the proposed methodology appears to be suitable for the investigation of a class of problems for which a solution could be difficult to be found while using other techniques. The proposed approach is capable of overcoming the limitations of other solution schemes thanks to the capability of FEM of solving linear and nonlinear boundary value problems with arbitrary material constitutive laws and geometries. Moreover, the use of the recently developed interface finite element [6–8] has further advantages. First of all, the possibility of taking into account arbitrary shapes for the indenting profile as analytical functions that are embedded into the interface element. The ability of simulating partial slip scenarios involving finite sliding of the indenter should also be mentioned.

Moreover, as a key advantage when compared to other models that are available in the literature that neglect the effect of Coulomb friction, focusing on viscoelastic dissipation only, here viscoelastic effects and frictional effects can be simultaneously investigated, since they are inherently coupled in the formulation. Neglecting interface tangential tractions, together with their related coupling affecting the distribution of normal tractions, could be reasonable when incompressibility conditions are approached. On the other hand, several evidences can be found that, as the Young's modulus of a viscoelastic material changes with time, so does the Poisson's ratio. Because the latter quantity

governs the coupling between normal and tangential tractions, a fully coupled model is worth study for fine precision engineering applications. As a final remark, the proposed interface finite element has the further advantage of being easily extended for taking thermal effects into account. These could be relevant not only for the analysis of temperature transfer across the interface, but also to simulate frictional heat generation, thus leading to a thermodynamically accurate model that is capable of investigating a wide class of realistic viscoelastic dissipative phenomena.

**Author Contributions:** Conceptualization, J.B. and M.P.; methodology, J.B. and M.P.; software, J.B.; validation, J.B.; formal analysis, J.B. and M.P.; investigation, J.B. and M.P.; resources, M.P.; data curation, J.B.; writing—original draft preparation, J.B.; writing—review and editing, M.P.; visualization, J.B.; supervision, M.P.; project administration, M.P.; funding acquisition, M.P. All authors have read and agreed to the published version of the manuscript.

**Funding:** This research received funding from the Italian Ministry of University and Research through the Research Project of National Interest (PRIN 2017) "XFAST-SIMS: Extra fast and accurate simulation of complex structural systems" (grant no. D68D19001260001).

**Conflicts of Interest:** The authors declare no conflict of interest.

## Appendix A. Model Validation

The proposed framework has been tested against a Hertz indentation problem for validation. The solution of the FEM simulation is compared with the analytical solution of the equivalent half-plane $2D$ contact problem, in terms of the integral of the interface normal and tangential tractions $P_z$ and $Q_x$, respectively, given the same imposed displacements history. A parabolic profile has been used as a first order approximation of a circular rigid cylinder with unitary radius $R_i$. The profile makes contact on the flat side of a linear elastic semi-disk with plane strain Young's modulus $E^* = 814.7\,\mathrm{Pa}$ and radius $R_d = 5R_i$, which simulates a half-plane. The load history includes two far-field displacements, imposed to the rigid profile. First, a normal displacement is applied, starting from zero and linearly increasing to a maximum value $\Delta_{z,0}/R_i = 1 \times 10^{-3}$, reached at time $t_0$, see the black line in Figure A1. The normal displacement is then held constant, and a harmonic tangential displacement is applied, which increases up to a maximum $f\Delta_{z,0}$, being $f = 0.2$ the coefficient of friction, and then makes a complete cycle, see the red line in Figure A1. Such a maximum value of horizontal displacement is chosen to cause the incipient sliding of the cylinder, and this is indeed what happens if the response of the system in terms of frictional reaction forces is analysed.

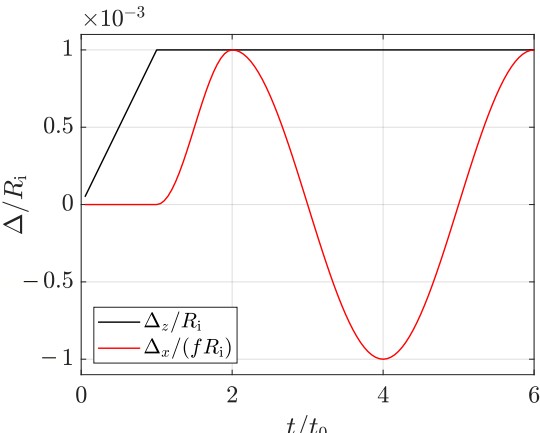

**Figure A1.** Imposed displacements.

*Appendix A.1. Evaluation of Normal Reaction Forces*

For plane contact problems, displacements can only be evaluated to within an arbitrary constant or, equivalently, in reference to a datum point. For the 2*D* Hertz problem, the boundary displacements normal to the interface can be evaluated as ([11], pp. 20–24):

$$w(x) = \begin{cases} -\frac{2P_z}{\pi E^*}\left[\left(\frac{x}{a}\right)^2 + c_0\right], \text{ if } x \le a, \\ -\frac{2P_z}{\pi E^*}\left[\log|\psi(x)| + \frac{1}{2\psi(x)^2} + \frac{1}{2} + c_0\right], \text{ if } x \ge a, \end{cases} \tag{A1}$$

where $c_0$ is the arbitrary constant, and:

$$\psi(x) = \frac{x}{a} + \sqrt{\left(\frac{x}{a}\right)^2 - 1}. \tag{A2}$$

An additional equilibrium equation relates the value of the load with the extension of the contact semi-strip *a*:

$$a = \sqrt{\frac{4P_z R_i}{\pi E^*}}. \tag{A3}$$

If the datum is set in correspondence of the point of the boundary $x = R_d$, the relation between the imposed displacement and the resultant vertical load has the form:

$$w(0) - w(R_d) = \Delta_z = \frac{2P_z}{\pi E^*}\left[\log \psi(R_d) + \frac{1}{2\psi(R_d)^2} + \frac{1}{2}\right], \tag{A4}$$

where $w(0)$ is evaluated in coincidence of the point of first contact, coincident with the centre of the semi-disk. As a final step, the inversion of Equation (A4) for a given value of $\Delta_z$ gives the desired $P_z$. The comparison with numerical results is shown in Figure A2, where diamond markers representing the FEM prediction show a very good accordance with the corresponding solid black line, that represents the analytical results.

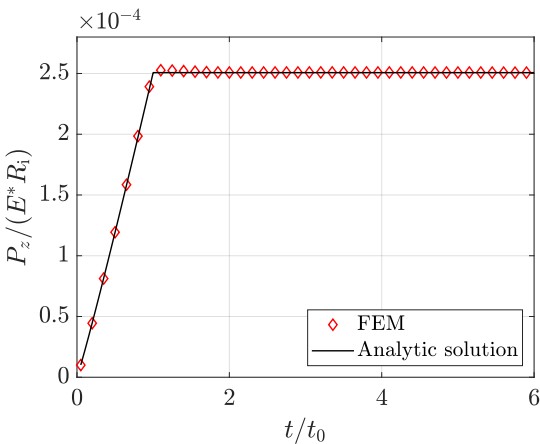

**Figure A2.** Resulting integrals of surface normal tractions.

*Appendix A.2. Evaluation of Tangential Reaction Forces*

Finding $Q_x$ for a given displacement still requires the evaluation of the applied displacement history with respect to a reference value, still set in correspondence of $x = R_d$. Since a closed form solution is not available for the tangential tractions, an extended version of the Jäger-Ciavarella theorem that accounts for variable normal and tangential loads have been used for evaluating the analytical solution of the problem, according to the algorithm presented in [12]. If a load path is defined in terms of $\Delta_z$ and $\Delta_x$, then, according to the theorem, the tangential problem can be reduced to the normal

one, since an increment in tangential forces can be evaluated as the difference between the actual vertical force and the vertical force related to a smaller imposed vertical displacement, multiplied by the coefficient of friction:

$$Q_x = f\left[P_z(\Delta_z) - P_z(\Delta_z^*)\right]. \tag{A5}$$

The value of $\Delta_z^*$ is a function of $\Delta_x$. For a constant normal load and an increasing tangential load, it can be evaluated as:

$$\Delta_z^* = \Delta_z - \frac{\Delta_x}{f}. \tag{A6}$$

For general loading scenarios, the principle can be extended and the correct value of $\Delta_z^*$ evaluated in terms of an equivalent path that respects both the equilibrium and the friction law. Results are shown in Figure A3,

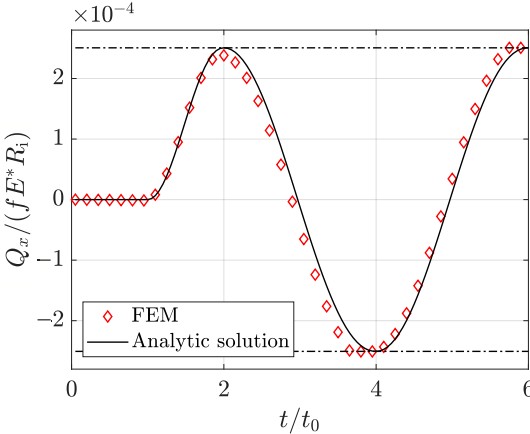

**Figure A3.** Resulting integrals of surface tangential tractions.

Where good accordance is found between the analytical solution given by the solid black line and the numerical prediction, depicted by the red diamond markers. In the same figure, the limit of *gross slip* for forward and backward sliding is shown as well by means of positive and negative valued horizontal black dash-dotted lines, respectively. These values represent the upper and lower threshold for the values of $Q_x$, and this condition is approached in correspondence of the related maximal tangential imposed displacement, cfr. Figure A1.

As a final remark, the differences between the numerical and the analytical results, for both normal and tangential forces, are due to the effect of coupling between normal and tangential tractions, which is not taken into account by the analytical approach. Moreover, another source of the small difference lies in the treatment of the friction law: FEM exploits a regularised Coulomb friction law, while the analytical approach exploits the classical one, where the stick-slip transition is abrupt.

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
