# Peer review of "Viscoelastic Effects during Tangential Contact Analyzed by a Novel Finite Element Approach with Embedded Interface Profiles"

_lubricants, doi:10.3390/lubricants8120107_

Round 1

Reviewer 1 Report

The authors proposed a novel finite element procedure for the frictional contact between a rigid indenter and a viscoelastic continuum. However, I have following concerns regarding the technical aspects of it.

(1) The abstract needs to include the main innovation and conclusions of this study.

(2) A brief introduction of the MPJR interface finite element method should be included in the model.

(3) All model assumptions and limitations must be stated clearly in the paper. For example, does the material exhibit linear viscoelastic property? Does the contact a plane stain problem or a plane stress problem?

(4) The authors should state the definition of the relaxation coefficients μn.

(5) The meaning of symbols  and Á in Figure 2 should be stated.

(6) Validations of the present model should be included.

(7) Is the normal displacement applied with a constant velocity or quasi-static? If it is applied with a constant velocity, the results at the end of the normal loading stage will be different for different velocities.

(8) The mesh is sparse and the results shown in Figure 3 are not accurate.

(9) The value of the maximum normal displacement, the sliding velocity and the friction coefficient should be given in section 4.

(10) From the present results, the advantage of MPJR interface finite element method over other methods cannot be found.

(11) The differences of input parameters between curves in Figure 4, for example the value of the applied normal displacement and the velocity of the horizontal displacement, should be given.

(12) Some grammatical errors exist in the whole manuscript. Examples are listed in the following. It should be noted that the errors are not limited to these. The authors should examine the whole manuscript carefully.

1) Lines 18: “and considerig a Maxwell rheological model”.

2) Lines 29: “Threfore, a simple model with a linear”.

3) Lines 95: “over the sperimental data”.

3) Lines 103: “Figuress 2b and 2c show the values of the loss”.

4) Lines 112: “Fourier transform of a power-law a power-law itself”.

5) Lines 115: “Threfore, relaxation times entering Prony series have to be regarder as design”.

6) Lines 129: “since coupling effect are fully included”.

Reviewer 2 Report

The authors presented an interesting improvement of the standard finite element method to deal with contact problems with any complex shape. They also provide an interesting example.

The overall presentation is very interesting and easy to follow. Just a remark: I know the authors provide several references, however I think it would be useful to better describe the MPJR interface finite element.

This is more a personal curiosity than a real comment: is this method applicable to 3D problems (with 2D contact surfaces)? If you think it can improve the paper, please comment.

Line 103: Figuress -> Figures

Line 181: "with a gap increasing from the single arm to the three arm model". From Fig. 5 it seems the opposite. Please check.

Fig. 5d: it seems that we are still far from the asymptote with three arms. How many arms do you think should be used to see a "converging" profile?

Round 2

Reviewer 1 Report

Comment (11) “The differences of input parameters between curves in Figure 4, for example the value of the applied normal displacement and the velocity of the horizontal displacement, should be given.” in my previous feedback is not addressed.

In section 2.3: The present problem is not the traditional elastic indentation problem which has the Hertz solution. Therefore, the analytical model which is used to validate the author’s framework should be cited and introduced simply. Furthermore, the material parameters used in the simulation should be presented.

The figures should be moved to the place after the paragraph that describes the corresponding figure.

There are still several grammatical errors. For example, the sentence ‘the Newton-Raphson iterative iterative method’ in Line 98 has two words ‘iterative’.
